# Onco-miR-21 Promotes Stat3-Dependent Gastric Cancer Progression

**DOI:** 10.3390/cancers14020264

**Published:** 2022-01-06

**Authors:** Janson Tse, Thomas Pierce, Annalisa L. E. Carli, Mariah G. Alorro, Stefan Thiem, Eric G. Marcusson, Matthias Ernst, Michael Buchert

**Affiliations:** 1Olivia Newton-John Cancer Research Institute, 145 Studley Road, Heidelberg 3084, Australia; janson.tse@gmail.com (J.T.); thomas_pierce@optusnet.com.au (T.P.); annalisa.carli@onjcri.org.au (A.L.E.C.); Mariah.alorro@gmail.com (M.G.A.); stethiem@gmail.com (S.T.); 2School of Cancer Medicine, La Trobe University, Melbourne 3083, Australia; 3Regulus Therapeutics, 10628 Science Center Drive, San Diego, CA 92121, USA; egmarcusson@gmail.com; 4Marcusson Consulting, 239 Brannan Street, San Francisco, CA 94107, USA

**Keywords:** miR-21, Stat3, EMT, TME, stomach tumor, antagomir

## Abstract

**Simple Summary:**

MicroRNAs (miRNAs) are a class of highly conserved small, non-protein coding RNAs with often-deregulated expression in cancer. miR-21 is a well-studied cancer-associated microRNA which is able to regulate proliferation, apoptosis, and invasion. The aim of this study was to investigate the molecular pathways which govern miR-21 expression in gastric cancer (GC), and to assess the therapeutic benefit of targeting miR-21 function with a small, synthetic miR-21 antagomir. We confirmed that miR-21 expression in a preclinical model of early gastric cancer is dependent on Stat3 downstream of the IL-6 family cytokine-mediated activation of gp130 receptor signaling. Antagomir therapy curbed gastric tumor growth, and restricted epithelial-to-mesenchymal transition and matrix remodeling. Our study established miR-21 as a promising anti-cancer target in GC.

**Abstract:**

MicroRNA-21 (miR-21) is a small, non-coding RNA overexpressed in gastric cancer and many other solid malignancies, where it exhibits both pro-and anti-tumourigenic properties. However, the pathways regulating miR-21 and the consequences of its inhibition in gastric cancer remain incompletely understood. By exploiting the spontaneous Stat3-dependent formation of inflammation-associated gastric tumors in *Gp130^F^**^/F^* mice, we functionally established miR-21 as a Stat3-controlled driver of tumor growth and progression. We reconciled our discoveries by identifying several conserved Stat3 binding motifs upstream of the miR-21 gene promoter, and showed that the systemic administration of a miR-21-specific antisense oligonucleotide antagomir reduced the established gastric tumor burden in *Gp130^F^**^/F^* mice. We molecularly delineated the therapeutic benefits of miR-21 inhibition with the functional restoration of PTEN in vitro and in vivo, alongside an attenuated epithelial-to-mesenchymal transition and the extracellular matrix remodeling phenotype of tumors. We corroborated our preclinical findings by correlating high STAT3 and miR-21 expression with the reduced survival probability of gastric cancer patients. Collectively, our results provide a molecular framework by which miR-21 mediates inflammation-associated gastric cancer progression, and establish miR-21 as a robust therapeutic target for solid malignancies characterized by excessive Stat3 activity.

## 1. Introduction

MicroRNAs (miRNAs) are short, non-coding, single-stranded nucleotides which bind to conserved 3′-untranslated region (3′-UTR) seed sequences in approximately 60% of all coding RNA to induce their rapid degradation or stall their translation [1]. Unlike conventional transcription factors, miRNAs provide the “fine tuning” of complex cellular responses in multicellular organisms, including proliferation, differentiation, cellular migration and others [2]. Furthermore, miRNAs are known to regulate immune responses and inflammation, and support regenerative processes and wound healing [3]. Therefore, many miRNA-dependent gene networks are commandeered by tumors and used in wound healing and inflammatory processes to support their progression [4]. Accordingly, tumor-promoting miRNAs (onco-miRs) can directly promote oncogenic activities, as well as impairing tumor suppressor functions. More recently, panels of miRNAs have shown promise as prognostic or diagnostic biomarkers for cancers of the breast [5], stomach [6], thyroid [7], colon [8], kidney [9] and lung [10].

miR-21 is one of the most prominent onco-miRs in most cancer types owing to its suppression of a network of *bona fide* tumor suppressor (TS) genes, including Reversion-inducing cysteine-rich protein with Kazal motifs (RECK), Tropomyosin 1 (TPM1), Phosphatase and tensin homolog (PTEN), Programmed Cell Death 4 (PDCD4) [11], Grainyhead-like transcription factor 3 (GRHL3) [12] and others. Akin to the genetic loss of function of mutations in TS genes, miRNAs cause the reduced expression of key TS proteins, which confers enhanced proliferation and survival to cancer cells, and promotes the invasive and metastatic properties of breast [13,14], colon [15], gastric [16] and other malignancies. Indeed, miR-21 is highly expressed in breast, ovarian, lung, prostate, pancreatic, colon and stomach cancers [11,17], and elevated miR-21 expression correlates with increased lymph node metastasis in cancer patients and reduced overall survival, [18,19] and the induction of chemoresistance [20,21,22,23]. Meanwhile, in preclinical models, enforced miR-21 expression induces malignant B-cell lymphoma, while the genetic silencing of miR-21 reduces tumorigenesis in transgenic mice [24]. Nevertheless, recent reports have demonstrated the anti-tumourigenic functions of miR-21 in liver and colon cancer models, highlighting the pleiotropic actions of this microRNA [25,26]. 

miR-21 expression is increased in response to the activation of the transcription factor NF-κB in bone marrow-derived macrophages [27], and by interleukin (IL)-6 and other inflammatory cytokines in myeloma cells [28]. Indeed, miR-21 expression has been linked to IL-6 signaling via the shared gp130 co-receptor subunit and the associated latent signal transducer and activator of transcription 3 (Stat3) [29]. Due to its critical role during wound healing and regeneration, Stat3 also plays a central role during the initiation, maintenance, and progression of solid cancers, both as a neoplastic cell intrinsic driver and as a negative modulator of the anti-tumor immune response [30]. Accordingly, increased Stat3 activity is also associated with lymph node metastasis and an overall poor prognosis in gastric cancer patients [31]. 

The therapeutic benefit of the targeting of miR-21 is suggested from preclinical observations in models of breast [32], lung [33] and colon cancer [34]. Given the strong link between elevated miR-21 expression and increased Stat3 activity, we hypothesized a functional involvement of miR-21 during the Stat3-dependent initiation, maintenance, and progression of gastric cancer [35]. Here, we exploit the *Gp130^F/F^* mouse as a pre-clinical model of Stat3-dependent gastric adenoma formation to delineate the contribution of miR-21 expression to early disease initiation and its underlying molecular mechanism. Importantly, we demonstrate that miR-21 is a Stat3-controlled *bona fide* onco-miR which presents a vulnerability that can be exploited by the therapeutic administration of an antisense-based miR-21-specifc oligonucleotide (“miR-21 antagomir”) which is currently being trialed in patients with Alport syndrome [36]. We reconcile these therapeutic benefits with the miR-21 antagomir-dependent re-expression of the tumor suppressor genes PTEN and PDCD4, alongside the repression of an epithelial-to-mesenchymal transition (EMT) and the associated extracellular matrix remodelling response.

## 2. Materials and Methods

### 2.1. Mouse Models and Treatments

The *Gp130^F^**^/F^*, *Pten^fl^*^/*fl*^, and the BAC-transgenic *gpA33:CreERT2* stains have previously been described [37,38,39], as has the *shStat3;rtTA* and *shLuc;rtTA* strain affording the doxycycline (DOX)-inducible expression of short hairpin (sh) RNA directed against Stat3 and firefly luciferase [40,41]. At 6 weeks of age, all *Gp130^F^**^/F^* mice develop gastric adenomas comprised of metaplastic transformed glandular epithelium [42]. 

Eight-week old mice received either anti-sense oligonucleotides (antagomirs) targeting miR-21 or a scrambled control (both from Regulus Therapeutics/Sanofi) at 15 mg/kg via intraperitoneal (i.p.) administration daily for a total of 4 weeks. The Stat3 expression in mice harboring the homozygous shStat3 allele was suppressed following the administration of DOX in food pellets (600 mg/kg, available ad libitum) for a period of 10 days. The *Pten* gene was conditionally deleted in the metaplastic gastric epithelium of *gpA33:CreERT2;Gp130^F/F^;Pten^fl/fl^* mice following the administration of tamoxifen (1 mg/mouse, i.p. twice daily for 5 consecutive days), and the tumors were collected 4 weeks after the last tamoxifen injection. 

Unless indicated, all of the mice were of an inbred C57BL/6 background propagated under specific pathogen-free conditions, and all of the experiments were approved by the LICR Institute’s and Austin Health’s Animal Ethics Welfare Committee. 

### 2.2. Antagomir Compounds

The Antagomir compounds were obtained from Regulus Therapeutics (San Diego, CA, USA), and were composed of 2′-O-methoxyethyl (2′MOE) and 2′-ribo-F-modified nucleotides on a phosphorothioate backbone. The miR-21 specific antagomir is complementary to the mature hsa-5P-miR-21 seed sequence (5′-UAGCUUAUCAGACUGAUGUUGA-3′), while the mismatch (MM) control contains a similar base content but lacks complementarity to the miR-21 seed sequence.

### 2.3. Cell Culture and Treatments

Human GC cell lines, AGS and MKN1 were acquired from the ATCC and tested for mycoplasma. All of the lines were maintained at 37 °C in 5% CO_2_, and were cultured in DMEM-F12 (Thermo Fisher Scientific, Waltham, MA, USA) supplemented with 10% FCS (Moregate Biotech, Bulimba, QLD, Australia ), 1% Penstrep (PS) (Thermo Fisher Scientific) and 1% Glutamax (Thermo Fisher Scientific). For the experiments, the cells were seeded at ~2.0 × 10^5^ cells/well density in 6-well plates overnight before stimulation with 1 µg/mL recombinant human IL-6, IL-11 (Resolving Images, AUS & NZ, Preston, VIC, Australia), or PBS (Thermo Fisher Scientific), and were harvested at the indicated time points. 

### 2.4. miRNA/mRNA Extraction and Q-RT-PCR

We used the mirVana miRNA Isolation kit (Thermo Fisher Scientific) to extract the miRNA from the tissues according to the manufacturer’s protocol, and micro-cDNA was synthesized using specific miRNA TaqMan primers (Appendix A) according to the manufacturer’s protocol. We used the secondary primer sets provided in the TaqMan miRNA kit (Thermo Fisher Scientific) to perform qRT-PCR analysis. We extracted mRNA from cultured cells using the QIAGEN mRNA extraction kit (QIAGEN, Clayton, VIC, Australia) according to the manufacturer’s protocol. In total, 1 microgram miRNA was used to synthesize the cDNA using a High-Capacity cDNA Reverse Transcription Kit (Applied Biosystems, Mulgrave, VIC, Australia) according to the manufacturer’s protocol. Q-RT-PCR was performed using the primers listed in Appendix A. The ∆∆CT method was used to calculate the relative miRNA or mRNA expression levels after normalization with housekeeping genes. The RNA quantity and quality were measured using the Nanodrop ND-2000 spectrophotometer (Nanodrop Technologies, Madison, WI, USA).

### 2.5. Chromatin Immuno-Precipitation 

Chromatin immunoprecipitation (ChIP) analyses were performed on gastric tumors isolated from *Gp130^F^**^/F^* mice 1 h after the i.p. administration of 5 µg recombinant human IL-6 or IL-11. DNA–protein cross-linking was performed by incubating the tissue in formaldehyde. The tissues were homogenized, the cells were lysed and the cell nuclei were collected. Following nuclear lysis, the chromatin was purified and sheared by sonication. ChIP reactions were performed using an Imprint ChIP kit (Sigma, Bayswater, VIC, Australia) using ChIP-grade Stat3 antibody (Santa Cruz #482X). Antibodies targeting RNA polymerase II and normal rabbit IgG and were used as the positive and negative controls, respectively. Next-generation sequencing (ChIP-seq), performed by Geneworks on an Illumina Genome Analyzer II platform, was used to obtain 35 bp sequencing reads from a single end of the ChIP-ed DNA fragments. The sequence reads were then mapped to the mouse genome to establish regions of enrichment (binding peaks).

### 2.6. Immunoblots

The tissues were homogenized in RIPA buffer using the TissueLyser II (QIAGEN) and centrifuged at 17,000× *g* to remove debris. The cells were collected by centrifugation at 300× *g* and lysed in RIPA buffer. The protein concentration was measured using the BCA method, as per the manufacturer’s protocol (Thermofisher). A total of 40 µg protein was resolved on a 4–12% Bis-Tris Gel (Invitrogen, Waltham, MA, USA) and transferred onto PVDF membranes via the iBLOT transfer system (Invitrogen). The blots were then probed with various antibodies overnight (Appendix A) and read using the LiCor Odyssey Imaging system (LiCor, Lincoln, NE, USA) after incubation with the appropriate fluorescent secondary antibody. Uncropped western blot scans are provided in Appendix A.

### 2.7. Lentiviral Packaging and Transduction

HEK293T cells were seeded 24 h prior to transfection in a 6-well plate (2 × 10^5^) in PS-free DMEM media. The DNA plasmids were diluted as follows: 15 µg pCMVdeltaR8.2 (lentiviral packaging plasmid), 6 µg pCMV-VSVG (encoding VSV envelope glycoprotein), 9 µg of either pEZX AM03 lentiviral miArrest miR-21 capture or pEZX AM03 lentiviral miArrest vector control expressing a synthetic scrambled control oligonucleotide (Genecopoeia) in 1.5 mL OPTI-MEM. The DNA was then combined with diluted Lipofectamine 2000 (65 µL in 1.5 mL OPTI-MEM). Transfections were performed for 48 h, after which the transfection media was removed and fresh PS free DMEM media was added. After 24 h, media containing the virus were collected, filtered using a 0.22 µm sterile syringe filter (MILLEX) and incubated with AGS or MKN1 cells cultured the night before. Cells incubated with the viral media were then subjected to spinoculation and incubated for 48 h before the media were replaced with fresh DMEM, 10% FCS. The stably transduced cells were isolated by flow cytometry using the mCherry marker. 

### 2.8. MTS Proliferation Assay

Cells were seeded into 96-well plates (5000 cells/well) and incubated overnight. The cultures were treated 24–72 h later with 20 µL MTS/PMS (Promega, Madison, WI, USA) solution and incubated for 90 min. The absorbance was measured at 490/630 nm wavelengths using a SPECTROstar Nano reader (BMG Labtech, Ortenberg, Germany) and was analyzed according to the manufacturer’s protocol (Promega). 

### 2.9. Migration and Invasion Assay

ThinCert™ Cell Culture 24-well inserts (Greiner Bio, Heidelberg West, VIC, Australia) were used to perform the migration and invasion assays. For the migration assays, 1 × 10^5^ cells per insert were seeded in serum-free media, and the inserts were placed in a 24-well plate well containing 300 µL DMEM-F12 + 10% FCS media, for 24 h. For the invasion assay, cells were seeded into an insert which contained 100 µL diluted serum-free Matrigel (1:10 dilution with serum free DMEM; BD Biosciences, Franklin Lakes, NJ, USA). After 72 h, the inserts were removed and fixed using 4% PFA for 10 min, washed, and stained with 1% Crystal Violet for 15 min. The inserts were then washed with PBS, and the inside was cleaned with a cotton swab before being photographed at 4× microscopic zoom. Fiji image analysis software was used for analyses. 

### 2.10. Clonogenic Growth Assay 

Clonogenic assays were performed by seeding 2 × 10^3^ cells into 20 × 100 mm dishes and culturing for 14 days. The plates were washed with PBS, fixed with 4% PFA for 10 min, washed, and stained with 1% Crystal Violet for 15 min. Digital images were taken of the entire 100 mm dish, and the number of colonies was enumerated using Fiji image analysis software.

### 2.11. Immuno-Histochemical Staining and Quantification 

The immuno-histochemical staining of the stomachs was performed as previously described [37], with slight alterations. The slides were incubated with primary antibodies overnight (Appendix A), washed, and then incubated with biotin secondary antibodies for 1 h. The signals were then amplified using the Vectastain Elite ABC HRP kit (Vector Laboratories, Meadowbrook, QLD, Australia). The signals were detected using HRP-conjugated secondary antibodies (Dako, Santa Clara, CA, USA) and the liquid diaminobenzidine (DAB) substrate chromogen system (Dako), before counterstaining with hematoxylin. The slides were scanned using the Aperio Digital Pathology Slide Scanner (Leica, Wetzlar, Germany), and the analyses were performed using scripts written on the Aperio system and Fiji image analysis software.

### 2.12. Survival Analysis

PTEN and PDCD4 survival plots were determined using a KM plotter, entailing gene expression data from *n* = 876 gastric cancer patients [43]. MiR-21 and STAT3 survival plots were determined based on gene expression data obtained from TCGA STAD datasets. The Z-score normalized expression for STAT3 and miR-21 were matched to the patient information for the OS event and OS time (days). Only tumor samples were used for the Kaplan Meier plots (*n* = 404). The gene expressions were median distributed (high > 0, low < 0) for each group. The individual KM plots were visualized in PRISM 8.4.3. 

### 2.13. Bioinformatics Data and Analysis

Pre-processed TCGA-STAD RNAseq Log2(RPKM+1), protein expression and miRNA mature strand expression data were downloaded via the UCSC Xena platform from the University of California, Santa Cruz (UCSC) [44], and were Z-score normalized per gene row (z = (x − μ)/σ). The clinical data—including tumor staging, Lauren classification and molecular subtype—were obtained from cbioportal [45,46] and linked to the TCGA-patient-ID number. Proteomics data miRNA expression data for the gastric cancer cell line were obtained from the CCLE platform [47,48]. The extracted data were used for Spearman-correlations performed in Rstudio version 4.0.0 (64 bit) [49].

### 2.14. Statistical Analysis 

A statistical analysis was carried out using GraphPad Prism VII software. The *p*-values were calculated by Student’s *t*-test, and were considered significant if *p* < 0.05. All of the data are expressed as the mean ± SEM. The statistical Log-rank (Mantel-Cox) test, Hazard ratios (logrank) and median survival were calculated in PRISM 8.4.3. 

## 3. Results

### 3.1. miR-21 Expression Is Elevated in the Stomach Adenoma of Gp130^F/F^ Mice, and Its Expression Is Dependent on Stat3 Signaling

Although overexpression studies have associated miR-21 expression with Stat3 signaling [28,29,50], it remains unknown whether a functional interaction between these molecules contributes to tumorigenesis in autochthonous cancer models. We therefore exploited the *Gp130^F^**^/F^* mouse model, which develops intestinal-type adenomas in the glandular stomach within the first two months through a strictly Stat3-dependent mechanism downstream of the shared interleukin (IL)-11 receptor gp130 [51]. The quantification of the abundance of mature miR-21-5p transcripts by TaqMan qRT-PCR revealed significantly higher miR-21 levels in adenomas than the surrounding unaffected tissue of the corpus and antrum of *Gp130^F/F^* mice (Figure 1B). In turn, the miR-21 expression in the latter tissues of *Gp130^F/F^* mice remained higher than in those collected from age-matched wild-type mice (Figure 1A,B), suggesting a direct correlation between Stat3 activity and miR-21 expression. 

In order to demonstrate that the increased expression of miR-21 was dependent of Stat3 signaling, we assessed the miR-21 expression in the stomachs of *Gp130^F/F^;Stat3^+/−^* mice, in which Stat3 expression is globally restrained and adenoma formation is blocked. Indeed, the restriction of the Stat3 signaling in *Gp130^F/F^;Stat3^+/−^* mice yielded similar miR-21 expression levels in the glandular antrum to those observed in the non-adenomatous antrum tissue of wild-type mice (Figure 1A,B), and the expression remained significantly lower than that in the unaffected glandular antrum and the adenomas of *Gp130^F/F^* mice. These observations extended as a trend into the corpus region, and demonstrated that miR-21 expression correlates with Stat3 expression and associated activation. 

In order to ensure that the increased expression of miR-21 observed in the *Gp130^F/F^* mice was due to increased Stat3 signaling compared to wild-type and *Gp130^F/F^;Stat3^+/−^* mice, Western blot analysis was performed to observe the levels of phosphorylated Stat3 (p-Stat3). As expected, the stomach tissues extracted from the *Gp130^F/F^* displayed the highest expression levels of p-Stat3 (Figure 1C). 

### 3.2. Stat3 Signaling Directly Induces miR-21 Expression 

In order to provide functional evidence for a Stat3 requirement of IL-11 dependent miR-21 induction in vivo, we exploited the *CAGsrtTA;Stat3.1348* (shStat3) mouse model, which enables the global suppression of Stat3 expression (and associated signaling) following the doxycycline (Dox)-dependent induction of a short hairpin RNA specific for Stat3 [41]. Whereas an IL-11 bolus challenge of Dox treatment-naïve shStat3 mice revealed the most significant and sustained induction of miR-21 expression in the liver, this effect was markedly blunted in the Dox-treated shStat3 mice (Figure 2A). These observations were replicated in stomach tissue extracted from these mice, where IL-11 administration only induced miR-21 expression in Dox treatment-naïve mice, but not in Dox treated shStat3 mice. We confirmed the specificity of this difference arising from reduced Stat3 expression by Western blot analysis, and observed a selective induction of the activated, i.e., tyrosine-705 phosphorylated, p-Stat3 isoform in response to IL-11 administration to Dox treatment-naïve shStat3 mice (Figure 2B). Collectively, these results argue strongly that miR-21 is a transcriptional target for Stat3 in gastric epithelium and the corresponding adenomas.

We next performed chromatin immunoprecipitation (ChIP) experiments to determine whether Stat3 binds directly to the miR-21 gene. The ChIP analysis of tumor DNA from *Gp130^F/F^* mice, prepared 60 min after a single IL-11 administration to activate Stat3 signaling, identified two potential Stat3 binding peaks at −38 kb and ~3.5 kb upstream of the miR-21 promoter and within introns of the overlapping transmembrane protein 49 (*TMEM49*) gene (Figure 2C). We substantiated a functional involvement of the identified Stat3 binding site for the regulation of miR-21 expression by assessing the interspecies conservation across the corresponding miR-21 sequence [52] for both binding peaks. Furthermore, the peak on mouse chromosome 11 at position 86437861 co-localized with the TTC(N_3_)GGA consensus sequence identified in the human miR-21 gene [53]. Moreover, we also detected this site in a separate Stat3 ChIP experiment on gastric tumor DNA of *Gp130^F/F^* mice following a single administration of the gp130-ligand IL-6. By contrast, an additional binding peak identified in response to IL-6 mapped >10 kb downstream of the miR-21 pre-mRNA sequence, and only conformed to a TTC(N_3_)GAA consensus sequence in the mouse genome.

### 3.3. Inhibition of miR-21 Attenuates Gastric Tumor Development Specifically in Stat3 Driven Gastric Tumors

In order to demonstrate that miR-21 expression is also subject to STAT3 signaling in human GC cells, we assessed the miR-21 expression in AGS and MKN1 cells stimulated with IL-6 and IL-11. In both cell lines, we observed a sustained induction of miR-21 over a period of 8–24 h (Figure 3A). We next used these cell lines to detect surrogate “anti-tumor” activity in vitro conferred by miR-21 inhibition [54], and used the miArrest^TM^ lentiviral system to generate the miR-21-5p knockdown GC cell lines AGS^miR−21KD^ and MKN1^miR−21KD^. The attenuation of miR-21-5p expression not only reduced the cellular proliferation significantly in cells grown in 2D on tissue culture plastic (Figure 3B) but also when grown in a physiologically more relevant manner as 3D colonies (Figure 3C). Because the inhibition of miR-21 has also been suggested to reduce EMT and metastasis [55], we extended our analysis of AGS^miR−21KD^ and MKN1^miR−21KD^ cells grown in Boyden chambers migration assays. We observed that miR-21 knockdown significantly impaired the migratory and invasive potential of these GC cell lines when compared to their miR-21 proficient non-targeting control lines (Figure 3D,E). In order to measure the impact of blunted miR-21 expression in the knockdown cell lines, we analyzed the protein expression of the *bona fide* miR-21 targets PTEN, BAX, and BCL2. As is consistent with previous observations [56], the PTEN and BAX expression was augmented upon miR-21 knock-down, whereas the expression of BCL2 was reduced in these cells (Figure 3F). Collectively, these data suggest that the suppression of miR-21 expression is likely to impair some of the tumor cell intrinsic cancer hallmarks, including proliferation and migration.

In order to further investigate the extent to which miR-21 contributed to Stat3-dependent gastric adenoma development, we first asked whether the treatment of adenoma-free *Gp130^F/F^* mice with a chemically-modified generation 2.5 miR-21 antisense oligonucleotide (“antagomir”) [57] could suppress de novo tumor formation (Figure 4A). We observed that administration of a miR-21 antagomir to 6-week-old *Gp130^F/F^* mice for 4 consecutive weeks significantly reduced the overall tumor burden when compared to a *Gp130^F/F^* littermate cohort treated with a scrambled antagomir of similar A/T nucleotide content. These observations were reproducible with mice of mixed C57Bl/6 × 129Sv, as well as an inbred C57Bl/6 background (Figure 4B,C and Appendix A). The reduced tumor burden in miR-21 antagomir-treated mice was underpinned by decreased Ki67 staining in the excised adenomas when compared to those from the scrambled antagomir control cohort (Figure 4D). As is consistent with a miR-21 dependent effect on cell proliferation, and hence tumor growth rather than tumor induction, we observed a decrease in the overall tumor burden rather than adenoma numbers (Figure 4C and Appendix A). We corroborated the systemic efficacy of the miR-21 antagomir after in vivo administration by confirming attenuated miR-21 expression in the adenomas and liver of the miR-21 antagomir-treated cohort when compared to tissue from mice treated with the scrambled antagomir (Figure 4E). Finally, in order to benchmark these observations against human GC, we correlated the expression levels of the mature miR-21-5p transcript against the recently reported functional pan-cancer “proliferation” gene expression signature [58] across the stomach cancer TCGA dataset, and found a significant positive correlation between them (Figure 4F). This suggests that miR-21 expression promotes proliferation across murine and human GC.

### 3.4. Stat3/miR-21-Dependent Pten and Pdcd4 Suppression Contributes to Gastric Tumorigenesis in Gp130^F/F^ Mice

In order to dissect the mechanisms by which miR-21 suppression limited the growth of Stat3-dependent gastric adenomas, we examined the expression of several candidate *bona fide* tumor suppressor genes that are regulated by Stat3 and miR-21. Specifically, the expression of the tumor suppressor genes *Pten* and *Pdcd4*, which are negative regulators of the pro-tumorigenic signaling pathways engaged by the Gp130 receptor pathway, was attenuated in adenomas after the Dox-induced expression of a luciferase control hairpin in *Gp130^F/F^*;*shLuc*;*rtTA* mice, as opposed to mice expressing a Stat3 hairpin which displayed the significant expression of both Pten and Pdcd4 (Figure 5A). Likewise, Pten and Pdcd4 protein expression was restored in the miR-21 antagomir-treated cohort of tumor-bearing *Gp130^F/F^* mice when compared to the scrambled antagomir-treated cohort (Figure 5B).

Given the significance of PTEN as a hallmark tumor suppressor in a majority of human malignancies, including GC, we next assessed the functional contribution of Pten to Stat3-driven adenoma formation in the *Gp130^F/F^* model. By exploiting the capacity of the transgenic *gpA33*:CreERT2 driver to restrict tamoxifen-inducible DNA-recombinase activity to the adenomatous epithelium in the stomach of *Gp130^F/F^* mice [37], we observed that the tamoxifen-induced deletion of Pten in 8-week old *gpA33:*CreERT2;*Gp130^F/F^*;*Pten*^fl/fl^ compound mice produced significantly larger adenomas 4 weeks later (Figure 5C). We confirmed the loss of Pten expression after tamoxifen administration in the tumor epithelium by the immunohistochemical staining of Pten using adenomas from the tamoxifen-induction naïve mice for comparison (Figure 5D). These observations support a pro-tumorigenic signaling cascade whereby Stat3 activation induces miR-21 to functionally mediate, in part, the tumor-promoting activity of excessive Stat3 by the suppression of Pten expression.

### 3.5. Inhibition of the Stat3-miR-21 Signaling Cascade Suppresses EMT and Promotes Fibrosis in GC

Having observed that the attenuation of miR-21 in vitro reduced the migration and invasion of GC cells, we next questioned whether miR-21 regulated EMT and the associated metastatic traits. We compared the RNA and protein expression of Vimentin (VIM), α-smooth muscle actin (ACTA2), and TWIST1 between the parental AGS and MKN cell lines and their miR-21 knock-down counterparts, and detected the attenuated expression of these EMT markers in the AGS^miR−21KD^ and MKN1^miR−21KD^ cell lines (Appendix A). Similarly, the immunohistochemical analysis of the adenomas excised from the miR-21 antagomir-treated cohort demonstrated the lower expression of these EMT markers, and correspondingly increased E-CADHERIN staining when compared to adenomas recovered from mice treated with the scrambled antagomir (Figure 6A). We also confirmed attenuated Vim and Acta2 expression in tumors from Dox-treated *Gp130^F/F^*;*shStat3*;*rtTA* mice compared to their control luciferase hairpin counterparts (Figure 6B). Finally, in order to benchmark these observations against human GC, we correlated the expression levels of the mature miR-21-5p transcript against the recently reported functional pan-cancer “EMT” gene expression signature [58] across the stomach cancer TCGA dataset, but found no significant correlation between them (Figure 6C). This suggests that the association between miR-21 and EMT is weak, and may be dependent on the cancer subtype and context; it is not conserved across murine and human GC.

Given the prevalence of the excessive extracellular matrix deposition and tissue fibrosis observed in GC, and the role of miR-21 in stimulating myofibroblast expansion and the associated augmentation of a fibrous matrix [59], we assessed the extracellular matrix deposition in the tumors of *Gp130^F/F^* mice. The trichrome and Picrosirius Red histochemical stains of tumors excised from miR-21 antagomir-treated *Gp130^F/F^* mice were less intense than the corresponding stains of tumors from the control-treated groups, suggesting a less-collagenous extracellular matrix (Figure 7A). Indeed, in addition to the reduced Acta2 staining observed in the tumors of miR-21 antagomir-treated mice (Figure 6A), we also noted reduced immunohistochemical signals for the two extracellular matrix components fibronectin and fibroblast activation protein alpha (FAP) (Figure 7B) when compared to adenomas extracted from scrambled antagomir-treated mice. Collectively, our data support the hypothesis that miR-21 expression promotes extracellular matrix remodeling in favor of a more fibrous tumor microenvironment, which can be attenuated with miR-21 inhibition. Again, finally, to benchmark these observations against human GC, we correlated the expression levels of the mature miR-21-5p transcript against the recently reported functional pan-cancer “matrix remodeling” gene expression signature [58] across the stomach cancer TCGA dataset, and found a highly significant positive correlation between them (Figure 7C). This suggests that miR-21 expression promotes extracellular matrix (ECM) remodeling across murine and human GC.

### 3.6. Expression of the Components of the Stat3/miR-21/EMT Signaling Cascade Correlates with GC Patients’ Survival

In order to further extend our findings to human GC, we next determined an association between STAT3/miR-21, their corresponding downstream targets (PTEN, PDCD4) and the survival outcomes in GC patients. Using TCGA and KMplotter databases, we found that patient cohorts with both elevated miR-21 and STAT3 expression were associated with poorer overall survival, while this association was inverted for the miR-21 targets PTEN and PDCD4 (Figure 8A,B). Interestingly, and in support of our results, only the cohort with high miR-21 and high STAT3 expression predicts a statistically significant worse outcome, while the high miR-21 and low STAT3 cohorts do not when compared to cohorts with low miR-21 expression, regardless of STAT3 levels. We also observed that miR-21 expression was consistently elevated in stomach cancer across all of the molecular and histological subtypes, and across the tumor stages when compared to the expression levels in healthy stomach controls (Figure 8C). Lastly, we prepared single correlations of the miR-21 transcript and protein levels of miR-21 targets PTEN and PDCD4 across all of the individual molecular and histological GC subtypes. This analysis returned a single significant negative correlation between miR-21 and PCDC4 in the intestinal subtype of GC, while all of the other correlations were not significant (Figure 8D and Appendix A). This is intriguing insofar as the transcriptomic profile of the gastric adenomas in the *Gp130^F/F^* mice is most closely related to the human intestinal-type GC [60], and thus further validates our findings.

Collectively, our results support the existence of a therapeutically targetable STAT3/miR-21 signaling cascade that partakes in gastric adenoma formation in mice through the impairment of the expression of tumor suppressors in favor of the genes involved in EMT and extracellular matrix production. The consistent association of these findings with the survival data in patients suggests that this cascade is likely to promote disease in humans.

## 4. Discussion

The last decade has firmly established that miRNAs can promote tumor progression in humans by conferring roles as oncogenes or tumor suppressor genes [11]. miR-21 is one of the most frequently overexpressed miRNAs in breast, ovarian, lung, prostate, pancreatic, colon, stomach, and many other solid malignancies [11,13,15,16,18,19]. Indeed, high miR-21 expression in gastric cancer patients is correlated with poor prognosis and lymph node metastasis [18,19]. Meanwhile, some of these “guilty-by-association” observations have been functionally corroborated. Specifically, Stat3—which in its own right confers many of the cell-intrinsic hallmarks of cancer—regulates miRNA expression, including miR-21 expression in cultured cells [28,29,50,61]. In order to investigate whether Stat3-dependent miR-21 regulation contributes to Stat3-dependent tumor formation, we exploited the *Gp130^F/F^* mouse as a clinically relevant model of intestinal-type gastric cancer. This enabled us to functionally map a pathway, whereby a majority of the tumor-cell-intrinsic cancer hallmark activities conferred by excessive Stat3 activity are in part mediated through miR-21 and the downstream suppression of tumor suppressor genes, as well as the promotion of an EMT conversion and extracellular matrix turnover.

The sequence analysis of the promoter region of the human miR-21 gene identified several conserved enhancer elements which contain potential binding sites for AP-1, C/EBP- α, p53 and Stat3 [53]. This includes three consensus TTC(N_3_)GAA Stat3 binding sites, which we confirm here by the ChIP analysis of DNA isolated from gastric tumors from *Gp130^F^**^/F^* mice stimulated with IL-11 and comprising the two functionally validated proximal sites at positions 86,400,807 and 86,400,717 in melanoma cells [28]. In addition, we observed Stat3 binding to a third, more distal TTC(N_3_)GAA site, which is perfectly conserved across the vertebrate kingdom. It remains to be established whether this site, due to its substantial distance from the transcriptional start site, may still act as an enhancer element through the formation of Stat3 tetramers, which are known to promote cooperation between distant DNA regions through the induction of DNA bending [62]. Additionally, our data do not exclude the involvement of other transcription factors in the regulation of miR-21 expression in gastric cancer.

The impairment of the expression of KLF5, RECK, TMP1, PDCD4, PTEN or other individual miR-21 target genes plays pivotal roles in enabling the progression of many solid cancers [11]. Given the multitude of genes regulated by miR-21, our hypothesis that miR-21 tunes entire gene networks to collectively support tumor-promoting characteristics is well supported. Among these, PTEN is the most widely implicated direct miR-21 target gene that enables miR-21 to function as an onco-miR [11,63]. Mechanistically, miR-21 suppresses the expression of PTEN, thereby counteracting the PI3K/AKT signaling cascade and the associated stimulation of cell survival, proliferation, migration and invasion [64]. Consistent with this, we observe here that both the antisense-mediated reduction of Stat3 expression in shStat3 mice and the antagomir-based suppression of miR-21 restore Pten expression [65], and the phenotypic consequences on these genetic manipulations on the tumor burden of *Gp130^F/F^* mice was opposite to that observed following the genetic ablation of *Pten* in these mice. Despite the strong association of miR-21 and Pten in the *Gp130^F/F^* mouse model, to our surprise, correlation studies using miR-21 transcript expression data and PTEN protein expression data from the TCGA STAD datasets produced no significant results across human GC tumor types. Potential explanations for this discrepancy are that the miR-21/PTEN regulatory axis may only be functional during early gastric tumorigenesis, or that PTEN is not a prime miR-21 target in human cancers. In contrast, the tumor suppressor PDCD4 turned out to be a much more consistent miR-21 target across mouse and human models. Future studies should further explore the functional role of the miR-21/PDCD4 axis in gastric cancer, for example by generating *PDCD4* CRISPR knockout cell lines or by genetically deleting *Pdcd4* in the *Gp130^F/F^* mouse model.

In addition to its role in the promotion of cellular migration and invasion, miR-21 also promotes metastatic processes by promoting EMT [55], and in particular the expression of VIMENTIN, SNAIL, ACTA2 and TWIST1 [66]. Furthermore, miR-21 downstream targets such as PTEN have also been shown to be involved in EMT [67]. Functionally, however, while we and others have demonstrated that the inhibition of miR-21 corresponds to the attenuation of several *bona fide* EMT markers, only two studies have functionally demonstrated that miR-21 promotes metastasis in vivo. Bornachea et al. demonstrated that miR-21 ablated transformed spindle cell carcinoma cell lines metastasized to the lung less when compared to the miR-21-WT cells [68]. Meanwhile, Li et al. demonstrated that GC-derived exosomal miR-21-5p induced mesothelial-to-mesenchymal transition (MMT) to provide a favorable environment for metastatic cancer cells [69]. However, whether miR-21 functionally induces metastasis in GC in vivo remains to be determined in models of more aggressive gastric cancer.

The increased turnover of the extracellular matrix is a well-documented driving factor in both cancer and fibrotic disease. Aside from providing vital structural supports in solid tumors, the ECM also mediates epithelial–stromal communication and regulates the flux of growth factors and chemokines, thus impacting tumor cell survival, proliferation, differentiation, metastasis and drug penetrance [70]. Increased miR-21 expression is vital in the development of fibrosis, as it promotes the proliferation of fibroblasts and increases the abnormal deposition of the extracellular matrix [59,71]. Our discovery that miR-21 inhibition reduces the fibrotic tumor microenvironment is reminiscent of findings that miR-21 inhibition attenuates the development and progression of fibrosis in cardiac and kidney tissue [72,73]. Additional work will be needed to detail the exact mechanisms of the ways in which miR-21 regulates fibrosis during tumourigenesis.

A major limitation of our current study was the use of the *Gp130^F/F^* mouse model, which only allowed us to study the impact of miR-21 inhibition on early gastric tumor development. Models of more advanced gastric tumorigenesis are needed to investigate the therapeutic benefit of miR-21 blockade on late-stage, metastatic disease.

Collectively, our study demonstrates that miR-21 provides an important signaling node downstream of Stat3 to confer tumor-cell-intrinsic cancer hallmarks, and therefore identifies miR-21 expression as a therapeutic cancer vulnerability. Given the broad activities of Stat3 as an enabler of cancer-cell-extrinsic hallmark capabilities, it will be important to establish the extent to which they also depend on miR-21 induction in the non-neoplastic cells of the tumor microenvironment.

## Figures and Tables

**Figure 1 cancers-14-00264-f001:**
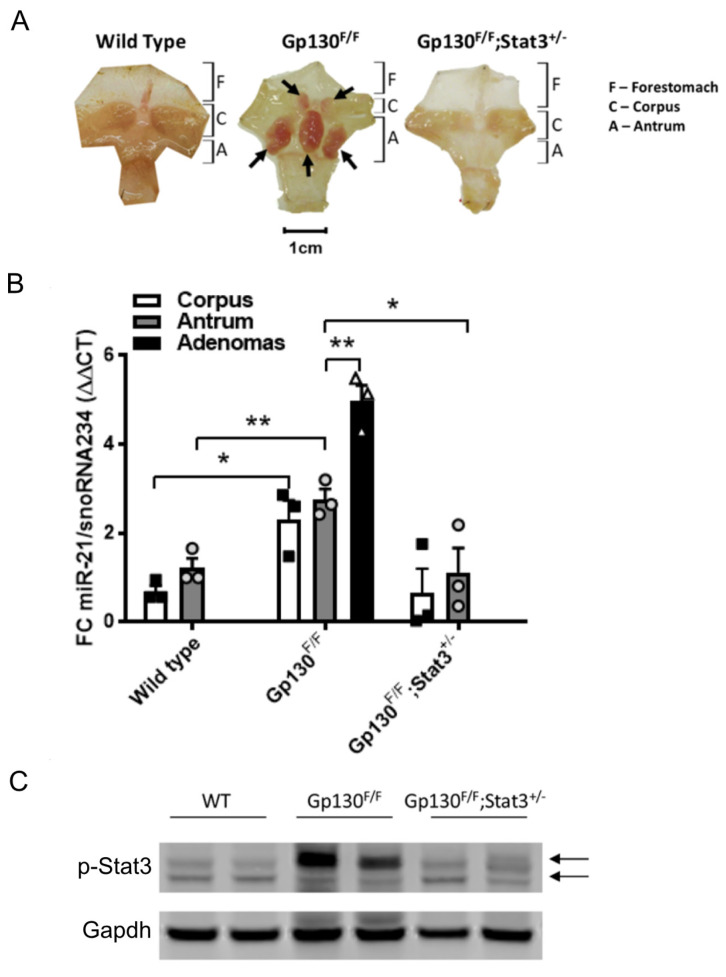
miR-21 expression in gastric tissues correlates with excessive Stat3 activity. (**A**) Whole-mount images of stomachs, excised from 8-week old mice of the indicated genotype and opened along the outer curvature. The arrows indicate adenomas in the glandular stomach. (**B**) qRT-PCR TaqMan analysis of the miR-21-5p expression in the unaffected corpus and antrum, as well as in gastric adenomas of 8-week old mice with the indicated genotypes. The individual circular, triangular and square data points represent the average values of technical triplicates normalized against the expression of snoRNA234. (**C**) p-Stat3 immunoblot analysis of excised stomach tissue. Gapdh was used as a loading control. Each lane corresponds to an individual mouse, with the arrows indicating the Stat3α and Stat3β isoforms. Each symbol represents an individual mouse, and the data are depicted as the mean ± SEM. *n* = 3 mice per group. * *p* < 0.05; ** *p* < 0.01.

**Figure 2 cancers-14-00264-f002:**
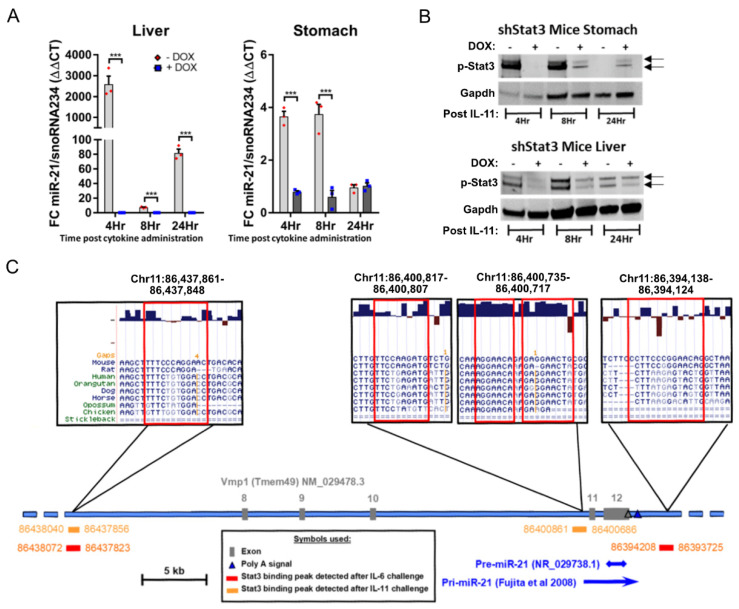
miR-21 is a direct transcriptional target of Stat3. (**A**) qRT-PCR TaqMan analysis of miR-21-5p expression in the liver and stomach of 6-week-old *Gp130^F/F^*;*shStat3*;*rtTA* mice injected with 5 µg IL-11 cytokine. Where indicated, the mice were fed doxycycline (DOX)-containing food pellets for the previous 5 days to induce shStat3 expression. Each symbol represents an individual mouse, and the data are depicted as the mean ± SEM. *n* = 3 mice per group, *** *p* < 0.001. (**B**) Western blot analysis for p-Stat3 in liver and stomach tissue extracted 4, 8 or 24 h after the i.p. administration of 5 µg IL-11 to *shStat3*;*rtTA* mice. Where indicated, the mice were fed DOX-containing food pellets, as in (**A**). Gapdh was used as a loading control. Each lane represents an individual mouse. (**C**) Representation of the murine Vmp1 gene locus containing the miR-21 gene on chromosome 11. As indicated in the boxed area above, there are three Stat3 binding peaks and a sequence near the pre-miR-21 start site identified by the Stat3 ChIP analysis performed on tumors isolated from 8-week-old *Gp130^F/F^* mice 1 hr after a single i.p. injection of 5 µg IL-6 or IL-11 to induce Stat3 signaling. The localizations of the IL-6 and IL-11-dependent Stat3 binding sites are indicated below the genomic locus in red and orange. The genome coordinates provided correspond to the C57Bl/6 mouse genome, NCBI build 37, 2007. The PhyloP conservation and genomic alignment data were retrieved using the UCSC Genome Browser (http://genome.ucsc.edu (accessed on 15 March 2020)).

**Figure 3 cancers-14-00264-f003:**
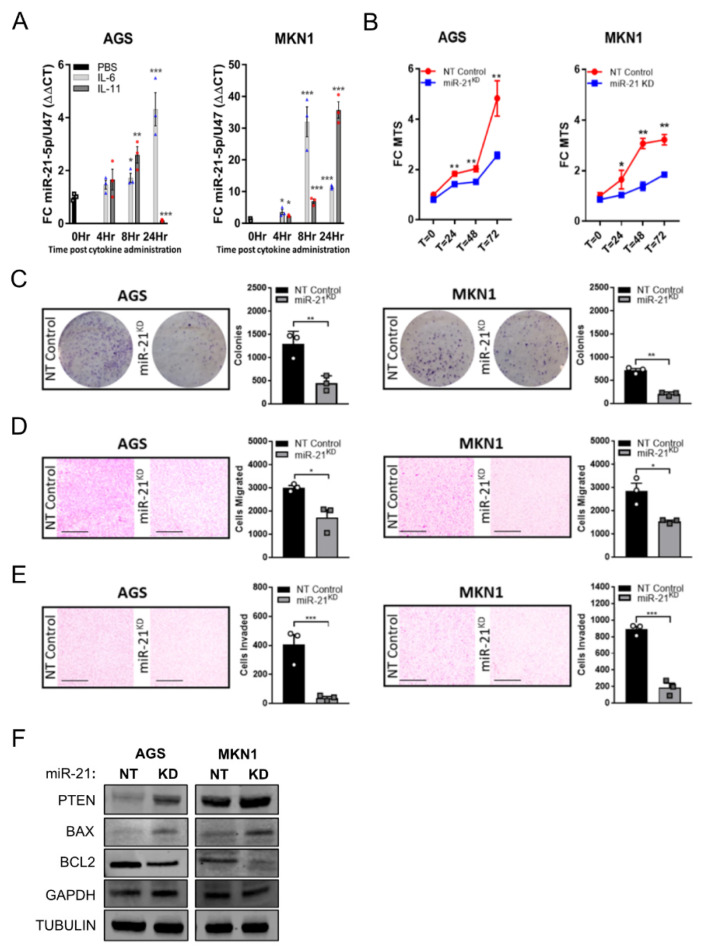
Inhibition of miR-21 attenuates the tumorigenic characteristics of human gastric cancer cells in vitro. (**A**) qRT-PCR TaqMan analysis of miR-21-5p expression in the human gastric cancer cell lines AGS and MKN1 following stimulation with IL-6 or IL-11 for 0, 4, 8 or 24 h. The individual circular, triangular and square data points represent the average values of technical triplicates normalized against the expression of the U47 housekeeping gene. (**B**) MTS proliferation assay of the human GC cell lines AGS and MKN1, either expressing an miR-21^KD^ construct or as their naïve (NT control) counterparts. The MTS-specific absorbance at 490 nm was expressed relative to the signal at the time of seeding the cells. (**C**) Clonogenic growth assays of AGS and MKN1 cells described in (**B**). The photographs show cell clones in a 100-mm diameter culture dish 10 days after seeding and stained with Crystal Violet. (**D**) Cell migration assays of AGS and MKN1 cells described in (**B**) with representative photographs of the lower side of a transwell 24 h after seeding. (**E**) Cell invasion assays of AGS and MKN1 cells described in (**B**) with presentative photographs of the lower side of a transwell 72 h after seeding. (**F**) Immunoblot analysis for miR-21 targets in AGS and MKN1 cells described in (**B**) using GAPDH and α-TUBULIN as loading controls. Each symbol represents a data point from one of *n* = 3 experiments, and the data are depicted as the mean ± SEM. * *p* < 0.05; ** *p* < 0.01; *** *p* < 0.001. Scale bar = 200 μm.

**Figure 4 cancers-14-00264-f004:**
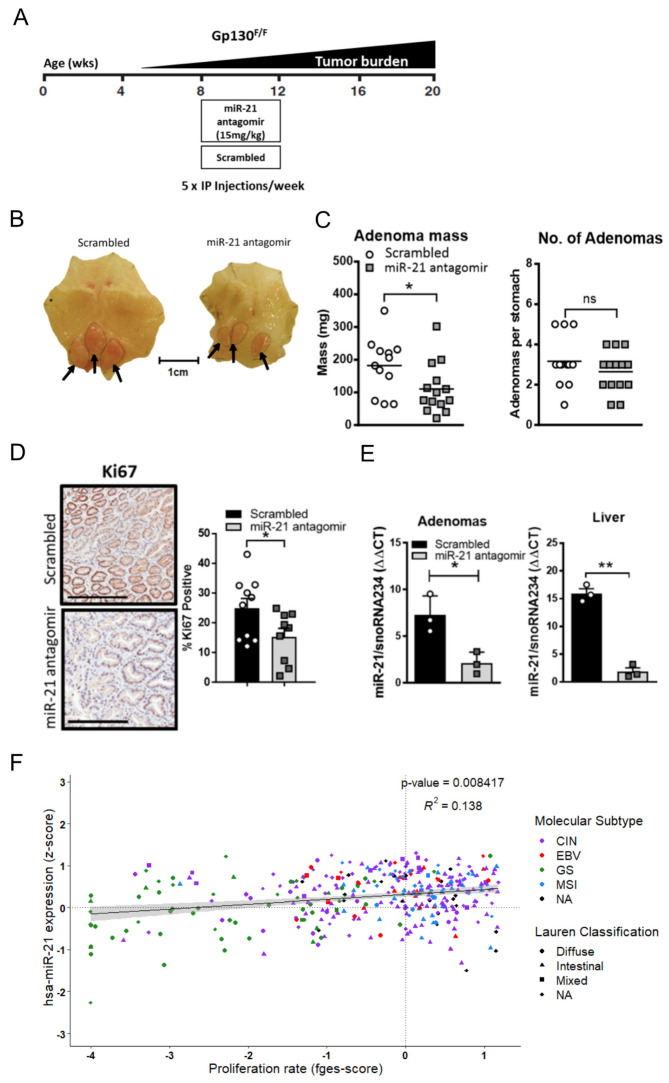
Pharmacological inhibition of miR-21 in vivo attenuates gastric adenoma growth. (**A**) Schematic of the treatment schedule of *Gp130^F/F^* mice. (**B**) Representative whole mounts of stomachs collected after the end of a 4-week treatment period with the scrambled of miR-21 antagomir that was commenced in 8-week old *Gp130^F/F^* mice on a C57Bl/6 × 129Sv background. The stomachs were opened along the outer curvature. The arrows indicate adenomas in the glandular part. (**C**) Enumeration of the total tumor mass and individual adenomas per *Gp130^F/F^* mouse treated as described in (**B**). (**D**) Ki67 immuno-histochemical staining of cross-sections of the adenomas from *Gp130^F/F^* mice treated as described in (**B**). The quantification of the Ki67 stains was carried out with ImageScope analysis software. (**E**) Q-RT-PCR TaqMan analysis of the miR-21 expression in the adenomas and livers of *Gp130^F/F^* mice treated as described in (**B**) and normalized against the expression of the housekeeping gene SnoRNA234. (**F**) Correlation analysis of miR-21 (z-score) with the “proliferation” functional gene signature (fges) using the TCGA STAD dataset. Each dot represents a single cancer, indicating both its molecular subtype and Lauren classification. Each symbol represents a data point from an individual mouse, and the data are depicted as the mean ± SEM. * *p* < 0.05; ** *p* < 0.01; n.s., not significant. Scale bar = 200 μm.

**Figure 5 cancers-14-00264-f005:**
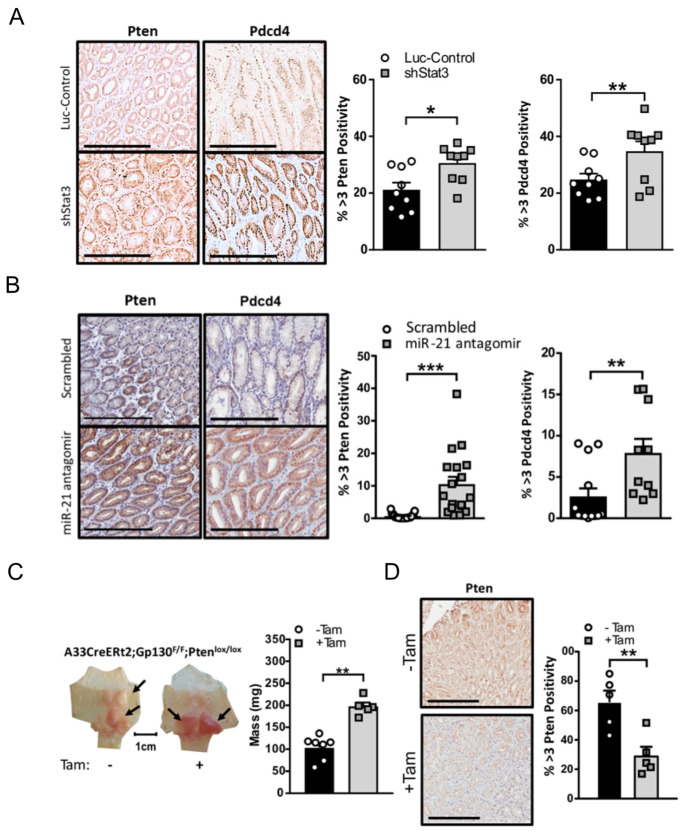
miR-21-dependent suppression of Pten and Pdcd4 contributes to gastric adenoma formation in mice. (**A**) Representative images of Pten and Pdcd4 IHC stains on cross-sections of adenomas from 18-week old *Gp130^F/F^*;*shStat3*;*rtTA* and *Gp130^F/F^*;*shLuc*;*rtTA* mice 2 weeks after DOX treatment. The quantification of the staining was performed by ImageScope analysis software. (**B**) Representative images of Pten and Pdcd4 IHC stains on cross-sections of adenomas collected from 12-week old *Gp130^F/F^* mice after the end of a 4-week treatment period with the scrambled control or miR-21 antagomir. The quantification of the staining was performed by ImageScope analysis software. (**C**) Representative whole mounts of stomachs excised from 12-week old *gpA33:*CreERT2;*Gp130^F/F^*;*Pten*^fl/fl^ mice 2 weeks after tamoxifen administration to ablate *Pten* in the the metaplastic epithelium of the tumors. Each symbol represents the total tumor weight of an individual mouse. (**D**) Representative images of Pten and Pdcd4 IHC stains in adenomas of mice from (**C**). The quantification of the staining was performed by ImageScope analysis software. The data are presented as the mean ± SEM, with each point representing an individual mouse. * *p* < 0.05; ** *p* < 0.01; *** *p* < 0.001. Scale bar = 200 μm.

**Figure 6 cancers-14-00264-f006:**
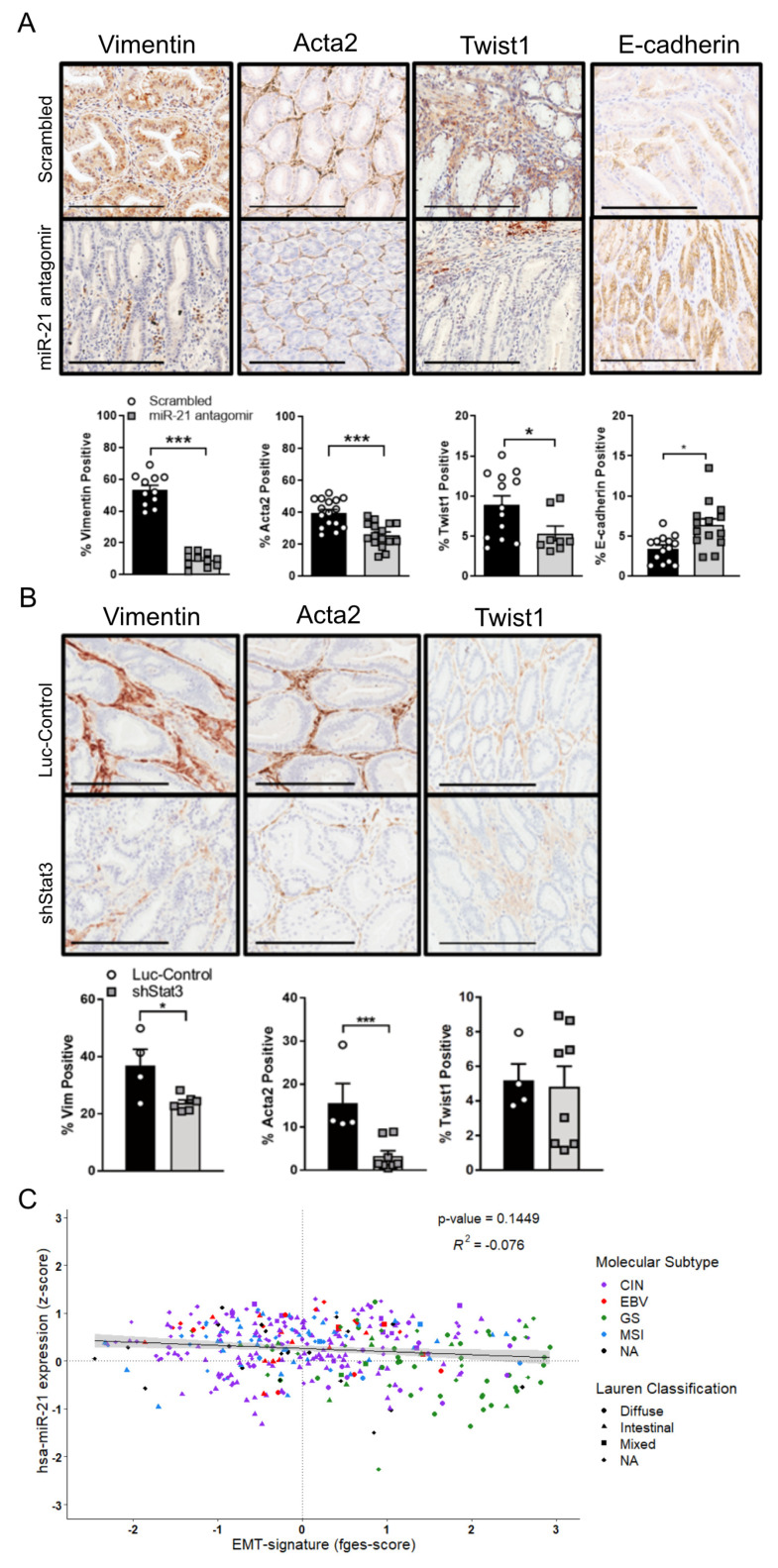
miR-21 inhibition suppresses the expression of EMT markers in Stat3-dependent GCs. (**A**) Representative IHC stains for Vimentin, Acta2, Twist1 and E-cadherin on cross-sections of adenomas collected in 12-week old *Gp130^F/F^* mice after the end of a 4-week treatment period with the scrambled control or miR-21 antagomir. The quantification of the staining was performed by ImageScope analysis software. (**B**) Representative IHC stains for Vimentin, Acta2 and Twist1 on cross-sections of adenomas from 18-week old *Gp130^F/F^*;*shStat3*;*rtTA* and *Gp130^F/F^*;*shLuc*;*rtTA* mice 2 weeks after DOX treatment. The quantification of the staining was performed using ImageScope analysis software. (**C**) Correlation analysis of miR-21 (z-score) with the “EMT-signature” functional gene signature (fges) using the TCGA STAD dataset. Each dot represents a single cancer, indicating both its molecular subtype and Lauren classification. The data are presented as the mean ± SEM, with each point representing an individual mouse. * *p* < 0.05; *** *p* < 0.001. Scale bar = 200 μm.

**Figure 7 cancers-14-00264-f007:**
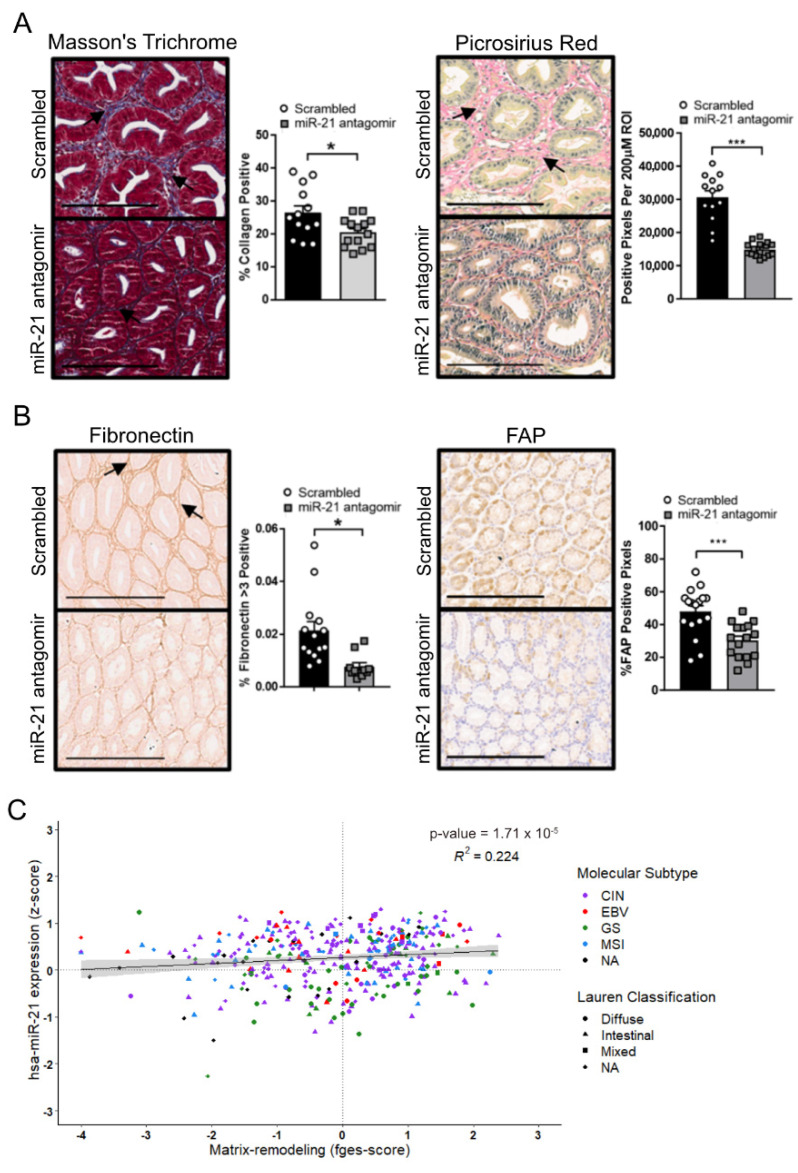
miR-21 promotes ECM remodeling in the tumors of *Gp130^F/F^* mice. (**A**) Representative *Masson’s* trichrome and *picrosirius red* stains on cross-sections of adenomas collected from 12-week old *Gp130^F/F^* mice after the end of a 4-week treatment period with the scrambled control or miR-21 antagomir. The quantification of the staining was performed by ImageScope analysis software. (**B**) Representative IHC stains for fibronectin and fibroblast-activated protein (FAP) on cross-sections of adenomas collected from 12-week old *Gp130^F/F^* mice after the end of a 4-week treatment period with the scrambled control or miR-21 antagomir. The quantification of the staining was performed by ImageScope analysis software. (**C**) Correlation analysis of miR-21 (z-score) with the “Matrix-remodeling” functional gene signature using the TCGA STAD dataset. Each dot represents a single cancer, indicating both its molecular subtype and Lauren classification. The data are presented as the mean ± SEM, with each point representing an individual mouse. * *p* < 0.05; *** *p* < 0.001. Scale bar = 200 μm.

**Figure 8 cancers-14-00264-f008:**
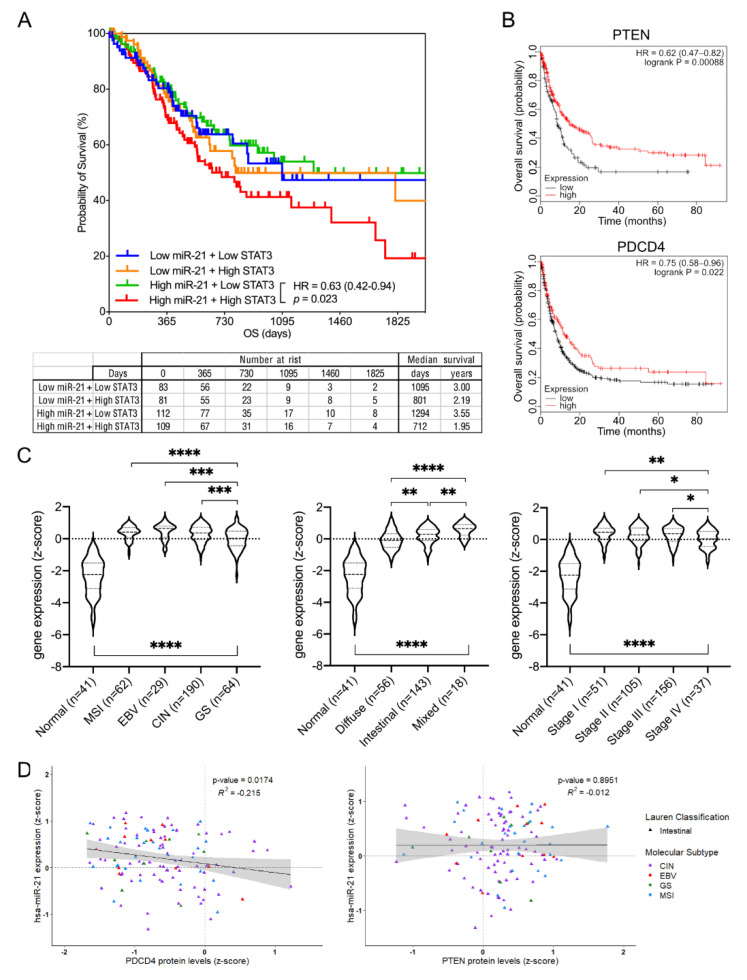
Transcriptomic and proteomic expression analyses across the Stat3-miR-21 signaling cascade in human GC. (**A**) Overall survival analysis of gastric cancer patient cohorts separated into the low and high expression of miR-21 and STAT3, respectively. (**B**) Overall survival analysis of gastric cancer patient cohorts separated at the mean expression into the low and high expression of the mir-21 target genes PTEN or PDCD4. (**C**) mir-21 transcript levels in the TCGA-STAD dataset grouped according to their molecular subtype (left), Lauren classification (middle) and tumor stage (right). (**D**) Correlation analysis of the mir-21 transcript and mir-21 targets PDCD4 and PTEN protein expression levels (z-score) in the Lauren intestinal-type GC using the TCGA STAD dataset. Each dot represents a single cancer, indicating both its molecular subtype and Lauren classification. * *p* < 0.05; ** *p* < 0.01; *** *p* < 0.001; **** *p* < 0.0001.

## Data Availability

The data presented in this study are available on request from the corresponding authors.

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
