# Peer review of "Onco-miR-21 Promotes Stat3-Dependent Gastric Cancer Progression"

_cancers, 2022, doi:10.3390/cancers14020264_

Round 1
Reviewer 1 Report
Tse and Pierce et al. produce a tremendous amount of data on the topic of gastric cancer and its control by a clearly demonstrated microRNA suspect – miR-21. Their experimentation from multiple angles reveals the convincing interplay between STAT3-driven gastric cancer, miR-21 and interleukin signaling. Their measured interpretation of the data and their openness towards discussing the study’s limiting scope are admirable.
Some minor comments:
- Line 35: What regulates miR-21 expression – STAT3? If not, miR-21 alone must be predictive of gastric cancer prognosis. In fact, it would be highly interesting to look at low STAT3 but high miR-21 cases.
- Figure 3G is not a strong correlation; also using such data from cancer cell lines is not a very convincing way to say something like this, given the few data points. The rest of the panels in the figure hold weight on their own in terms of supporting the assertion that miR-21 is cancer promoting and that attenuation of the miRNA hampers cancer progression. I suggest removing panel G.
- While the experimental manipulations overall yielded coherent evidence for the causal role of miR-21 in GCs, the correlation panels from the TCGA and CCLE datasets (Figures 4F, 6C, 7C, 8D) were much less convincing. Could miR-21 only play an important role in STAT3-driven cancers and could this be the reason?
- In figure 3, it is not clear if the miR-21 knockdown experiments were controlled using a scrambled lentiviral treatment as negative control. In the absence of such a control, the effects observed cannot be solely attributed to the knockdown of miR-21.
- Line 431: Change “tamoxifen treatment” to “tamoxifen administration”.
Author Response
We thank the reviewer for the positive and considerate feedback. We have attempted to address each query to the best of our abilities. Below follows a point-by point summary:
Reviewer 1:
1. Line 35: What regulates miR-21 expression – STAT3? If not, miR-21 alone must be predictive of gastric cancer prognosis. In fact, it would be highly interesting to look at low STAT3 but high miR-21 cases.
We thank the reviewer for this comment. While miR-21 is predictive for relapse-free survival (previous Figure 8A), it is not predictive for overall survival (updated Figure 8A). Meanwhile high STAT3 predicts overall worse survival (previous Figure 8A). This suggests that miR-21 expression in gastric cancer is not solely controlled by STAT3, in fact earlier reports have shown regulation of miR-21 by NF-kB and AP-1 transcription factors (doi, 10.1158/0008-5472.CAN-10-2579; doi, 10.1186/s12935-019-0931-x). On the other hand, miR-21 is only one of several critical STAT3 targets in gastric cancer. The investigation of low STAT3/high miR-21 GC cases would undoubtedly be interesting but was beyond the scope of this study. Nevertheless we have now modified Figure 8A to show the impact of high/low miR-21 and STAT3 cohorts on overall survival in the TCGA STAD datasets. Interestingly, and in support of our results, only the cohort with high miR-21 and high STAT3 expression predicts a statistically significant worse outcome, while high miR-21 and low STAT3 cohorts do not when compared to cohorts with low miR-21 expression, regardless of STAT3 levels.
2. Figure 3G is not a strong correlation; also using such data from cancer cell lines is not a very convincing way to say something like this, given the few data points. The rest of the panels in the figure hold weight on their own in terms of supporting the assertion that miR-21 is cancer promoting and that attenuation of the miRNA hampers cancer progression. I suggest removing panel G.
We agree and have now removed this panel.
3. While the experimental manipulations overall yielded coherent evidence for the causal role of miR-21 in GCs, the correlation panels from the TCGA and CCLE datasets (Figures 4F, 6C, 7C, 8D) were much less convincing. Could miR-21 only play an important role in STAT3-driven cancers and could this be the reason?
We thank the reviewer for this comment. We believe that the TCGA and CCLE datasets better reflect tumour heterogeneity than our preclinical in vitro and in vivo models and therefore we felt it important to include data derived from these datasets as well to give our data a larger perspective.
4. In figure 3, it is not clear if the miR-21 knockdown experiments were controlled using a scrambled lentiviral treatment as negative control. In the absence of such a control, the effects observed cannot be solely attributed to the knockdown of miR-21.
We apologise for the confusion. The NT control describes a non-targeting control expressing a synthetic scrambled control oligonucleotide. We have now made this clear in the methods section (line 176).
5. Line 431: Change “tamoxifen treatment” to “tamoxifen administration”.
We have now changed description as requested (lines 455, 468).

Reviewer 2 Report
In this study Tse et al. demonstrate that Onco-miR21 drives Stat3-dependent gastric cancer progression molecularly contributing to PTEN and PCDC4 suppression and inducing EMT, ECM remodeling and fibrosis. Expression of Stat3/miR-21 signaling cascade components correlates a poorer relapse-free and overall survival in GC patients establishing miR-21 as robust therapeutic target for stomach tumor. The manuscript is well written and clear and the results provided are sound (the results are supported by solid in vitro and in vivo experiments taking advantage also of several elegant transgenic mouse models). Only minor comments could be raised:
FIG 2C: Enlarge
FIG 3F: Show, if , possible, also PDCD4 western blot
FIG 3G: Show if possible the correlation also with PTEN, BAX and BCL2
Author Response
We thank the reviewer for the positive and considerate feedback. We have attempted to address each query to the best of our abilities. Below follows a point-by point summary:
Reviewer 2
FIG 2C: Enlarge
We have enlarged Fig 2C as requested
FIG 3F: Show, if , possible, also PDCD4 western blot
Unfortunately, while our PDCD4 antibody worked well for staining of formalin fixed and paraffin-embedded tissue sections, we were unable to generate convincing western blot data. In addition, largely based on comments by reviewer 1 who requested panel G in Figure 3 to be removed, we have now completely omitted PDCD4 data from this figure. However, PDCD4 data is still included as part of figure 5 and 8.
FIG 3G: Show if possible the correlation also with PTEN, BAX and BCL2
As explained above, the PDCD4 correlation of Figure 3 was removed from the manuscript and hence we choose not to include this type of analysis for PTEN, BAX and BCL2.

Reviewer 3 Report
The present focuses on miR and STAT3 interaction in inflammation and its association with progression of gastric cancer. Following revisions are required before final publication of manuscript:
- Title need to be changed and authors should provide more attractive title.
- The authors should be careful about writing statements. For instance, first sentence of abstract is completely wrong. Authors have mentioned that miR-21 is an oncogenic factor. The miRs have both functions and they act as double-edged sword. Look at these articles (Doi,10.3390/cancers13194983; Doi, 10.3389/fonc.2020.610545; and Doi, 10.1371/journal.pone.0260327) and you will see that miR-21 can function also as tumor-suppressor factor. Therefore, revise first statement of abstract and ba careful in other sections of manuscript.
- Micro-RNA (miRNA) are. It should be microRNAs (miRNAs) are. Change it and all sections of manuscript should be checked in terms of spell and grammar mistakes.
- 3’-UTR. Mention its full name for first time it appears in text.
- pro-liferation. Should be proliferation.
- The introduction needs re-organization. Authors have only discussed about miRNAs and miR-21. What about Stat3? In your introduction, you should focus on miRNAs, Stat3 and inflammation.
- The conclusion section should be elaborated by providing limitations of current work and adding more directions for future studies.
- A newly published article in Cancer letters journal can help you in improving introduction section about miRNAs and inflamamtion (Doi,10.1016/j.canlet.2021.03.025).
- There are only two references from 2020 and authors should update their information and refs to increase visibility of their article and to improve its quality.
Author Response
We thank the reviewer for the positive and considerate feedback. We have attempted to address each query to the best of our abilities. Below follows a point-by point summary:
Reviewer 3
1. Title need to be changed and authors should provide more attractive title.
We have now changed the title to “Onco-miR21 promotes Stat3-dependent gastric cancer progression”
2. The authors should be careful about writing statements. For instance, first sentence of abstract is completely wrong. Authors have mentioned that miR-21 is an oncogenic factor. The miRs have both functions and they act as double-edged sword. Look at these articles (Doi,10.3390/cancers13194983; Doi, 10.3389/fonc.2020.610545; and Doi, 10.1371/journal.pone.0260327) and you will see that miR-21 can function also as tumor-suppressor factor. Therefore, revise first statement of abstract and ba careful in other sections of manuscript.
We thank the reviewer for this comment. We have now revised the statement in the abstract, simple summary section and other relevant sections in the manuscript to reflect both the pro- and anti-tumourigenic properties of miR-21. See lines 16-17, 24-25, 63, and 75-78.
3. Micro-RNA (miRNA) are. It should be microRNAs (miRNAs) are. Change it and all sections of manuscript should be checked in terms of spell and grammar mistakes.
We have made the changes as requested and re-checked the manuscript for spell and grammar mistakes.
4. 3’-UTR. Mention its full name for first time it appears in text.
We have made the change as requested (line 45).
5. pro-liferation. Should be proliferation.
This has been corrected as requested (line 17).
6. The introduction needs re-organization. Authors have only discussed about miRNAs and miR-21. What about Stat3? In your introduction, you should focus on miRNAs, Stat3 and inflammation.
We thank the reviewer for this comment. Nevertheless, we respectfully disagree with this assessment. Our introduction already has an entire paragraph on microRNAs (line 45-55) and Stat3 features prominently throughout the introduction (lines 73-76 and 81-89). A large proportion of the introduction is dedicated to miR-21 and we believe this is justified because the entire manuscript is dedicated to the elucidation of the molecular mechanism of miR-21 and therapeutic benefit of targeting this microRNA. Inflammation is not the main topic of this study and therefore does not warrant an entire paragraph. We have now nevertheless included inflammation in the first paragraph as an additional example of a biological function that can be regulated by microRNAs (line 51). Furthermore, in an effort to improve quality and visibility of the article, we have now included several more recently published studies in the introduction (see also point 9).
7. The conclusion section should be elaborated by providing limitations of current work and adding more directions for future studies.
We thank the reviewer for the comments. We have now included additional sentences explaining limitations and suggested future studies. See lines 640-642, 665-667, 679-680, 690-691.
8. A newly published article in Cancer letters journal can help you in improving introduction section about miRNAs and inflammation (Doi,10.1016/j.canlet.2021.03.025).
We thank the reviewer for highlighting this article. The changes made in the introduction are elaborated above under point 6.
9. There are only two references from 2020 and authors should update their information and refs to increase visibility of their article and to improve its quality.
We thank the reviewer for this suggestion: We have now updated the reference list with 15 recent articles from 2021 (13) and 2020 (2). The references in question are 5, 6, 7, 8, 9, 10, 14, 17, 20, 21, 22, 23, 25, 26, and 71.
